# Meta-analysis of the efficacy and safety of Ginkgolide Meglumine Injection combined with Butylphthalide in the treatment of Acute Ischemic Stroke

**Li Xin-Shuai**[1], **Jia-Qi Zhou**[1], **Xiang-Fan Chen**[2], **Xia Chen**[1]*, **Pan-Feng Feng**[1]*

**1** Department of Pharmacy, Affiliated Hospital 2 of Nantong University, First People's Hospital of Nantong City, Nantong, Jiangsu Province, China, **2** BioBank, Affiliated Hospital 2 of Nantong University and First People's Hospital of Nantong City, Nantong, Jiangsu Province, China

☯ These authors contributed equally to this work.
* cxia66@126.com (XC); 929083891@qq.com (P-FF)

## Abstract

### Objective

To evaluate the efficacy and safety of Ginkgolide Meglumine Injection (GMI) combined with Butylphthalide in the treatment of Acute Ischemic Stroke (AIS), and provide reference for rational clinical medication.

### Methods

PubMed, Embase, Web of science, CNKI, Wanfang, VIP and other databases were searched for published studies on the treatment of AIS with GMI combined with Butylphthalide in both Chinese and English. The search period was from the establishment of the database to July 2023. The included studies that met the inclusion criteria were analyzed using RevMan 5.3 software for Meta-analysis.

### Results

A total of 25 studies involving 2362 patients (experimental group = 1182, control group = 1180) were included. The results of meta-analysis showed that the overall effective rate of the experimental group was significantly higher than that of the control group [RR = 1.21, 95% CI (1.16, 1.26), P< 0.00001]. In addition, compared with the control group, the experimental group showed significant improvement in NIHSS score [SMD = -1.59, % CI (-2.00–1.18), P< 0.00001] and ADL score [SMD = 2.12, 95% CI (1.52, -2.72), P<0.00001], significant decrease in CRP [SMD = -2.24, 95% CI (-3.31, -1.18), P<0.0001] and TNF-α [SMD = -2.74, 95% CI (-4.45, -1.03), P< 0.005] levels, and improvement in plasma viscosity [SMD = -0.86, 95% CI (-1.07, -0.66), P< 0.00001]. However, the influence on homocysteine level remains inconclusive. Furthermore, there was no significant difference in the incidence of adverse reactions between the two groups [SMD = 0.95, 95% CI (0.71, 1.28), P> 0.05].

**Data Availability Statement:** All relevant data are within the paper and its Supporting Information files.

**Funding:** This work was supported by Fund of Nantong university (No.2022JZ005, 2022JY004), Fund of drug policy and pharmaceutical care of Nantong City (No.2023NTPA05), Jiangsu Pharmaceutical Association-HengRui Hospital Pharmacy Fund (No. H202047), Nantong Health Commission Fund (No. QA2021007, QA2021006, QA2021014) and Development Fund of KangDa college of Nanjing medical university (No. KD2021KYJJZD127). The funders had no role in study design, data collection and analysis, decision to publish, or preparation of the manuscript. There was no additional external funding received for this study.

**Competing interests:** The authors have declared that no competing interests exist.

## Conclusion

GMI combined with Butylphthalide shows good clinical application effects and good safety in the treatment of AIS. However, more large-sample, multicenter, randomized controlled are needed to confirm these findings.

## 1. Introduction

A study based on the 2019 Global Burden of Disease database revealed that, as of 2019, there were a total of 12.2 million new cases of stroke globally, with 101 million prevalent cases and a stroke-related Disability-Adjusted Life Years (DALYs) of 143 million. Stroke was responsible for the death of 6.55 million individuals. From 1990 to 2019, the absolute number of new stroke cases increased by 70.0%, the number of current stroke cases increased by 85.0%, the number of deaths increased by 43.0%, and the number of DALYs caused by stroke increased by 32.0% [1]. Globally, stroke remains the second leading cause of death (11.6% of total deaths), the third leading cause of death and disability (5.7% of total deaths). Over the past 30 years, as one of the main subtypes of stroke, the number of deaths from ischemic stroke has increased from 2.04 million to 3.29 million, and it is projected to reach 4.9 million by 2030 [1,2]. This has resulted in a significant economic burden and has become a severe global public health issue.

Acute ischemic stroke (AIS) is characterized by cerebral tissue damage caused by interruption of cerebral blood supply, primarily due to atherosclerosis, thrombosis, and embolism [3]. It has a rapid onset, usually occurring suddenly, and is clinically manifested by neurological deficits such as limb weakness, speech disorders, and sensory impairments [1–3]. Currently, commonly used treatment measures in clinical practice for AIS include thrombolysis, antiplatelet therapy, and symptomatic supportive treatment, which can partially restore blood flow to the ischemic area [3,4]. Additionally, studies have shown that the application of neuroprotective agents can reduce neuronal cell death and improve neurological function in patients [4].

Butylphthalide, chemically known as racemic 3-phenylbutan-2-one, is a newly developed chemical drug in recent years in China for the treatment of AIS. It has the characteristics of high lipid solubility, easy passage through the blood-brain barrier, direct action on the infarct site, rapid onset, and significant effects [5]. The "Guidelines for the Diagnosis and Treatment of Acute Ischemic Stroke in China 2018" recommend its use for improving neurological deficits in individualized treatment of mild to moderate AIS patients (Class II recommendation, Level B evidence) [6]. Multiple studies have shown that Butylphthalide inhibits neuronal apoptosis by blocking the cascade reaction of caspase [7]. In addition, Butylphthalide significantly improves neuronal cell function by protecting mitochondria, alleviating inflammatory response, and enhancing microcirculation, with no serious adverse reactions [5]. It has become one of the main therapeutic drugs for improving neurological deficits in mild to moderate AIS patients in China.

Ginkgolide Meglumine Injection (GMI) is a traditional Chinese medicine preparation extracted from Ginkgo biloba leaves using modern analytical methods. Its main active component, ginkgolide, is composed of ginkgolide K, B, A, etc. It has been used in China for the treatment of mild to moderate ischemic stroke, showing certain efficacy [8]. Modern pharmacological studies have shown that GMI, as a platelet-activating factor receptor antagonist, can effectively antagonize platelet aggregation, improve brain edema in AIS patients, protect neuronal cells, and promote neural function recovery [9]. Additionally, ginkgolide B and

A can protect brain neuronal cells by inhibiting the mitochondrial apoptotic pathway, reducing NF-κB activity, and alleviating neuronal apoptosis and inflammatory reactions [10]. The "China Acute Ischemic Diagnosis and Treatment Guidelines 2018" have suggested that Chinese medicine can improve neurological deficits in ischemic stroke and warrant further high-quality research for confirmation [6].

Although some studies have suggested that the combination of GMI and Butylphthalide may have higher clinical benefits in the treatment of AIS, the quality of these studies is uneven, the sample size of the studies is small, the efficacy indicators of the studies are few, and the reports on the research results are inconsistent. There is currently no systematic analysis of these small sample clinical trials. Therefore, we conducted this meta-analysis to comprehensively evaluate the efficacy and safety of GMI combined with Butylphthalide in the treatment of AIS, aiming to provide references for rational clinical medication.

## 2. Materials and methods

### 2.1 Study design

According to the Cochrane Handbook criteria and the Preferred Reporting Items for Systematic Reviews and Meta-Analyses (PRISMA), our study has been registered in PROSPERO (CRD42023445014). This study adheres to the PRISMA guidelines.

### 2.2 Search strategy

Researchers conducted literature searches in PubMed, Embase, Web of Science, China National Knowledge Infrastructure (CNKI), Wanfang Database (Wanfang), and VIP Database for Chinese Technical Periodicals (VIP) using the search terms "Ginkgolide Meglumine" "Butylphthalide" and "Acute Ischemic Stroke". All studies published in full text were eligible for inclusion, with no language restrictions. The search period ranged from the establishment date of each database to July 2023.

### 2.3 Inclusion and exclusion criteria

**2.3.1 Inclusion criteria.** All randomized controlled trials, cohort studies, or case-control studies, regardless of blinding were included in this study. All patients met the World Health Organization's diagnostic criteria for AIS. The control group received treatment with Butylphthalide or GMI alone, while the experimental group received combined treatment with GMI and Butylphthalide. The primary outcome measure was efficacy rate. Secondary outcome measures included the NIHSS score (i.e. National Institutes of Health Stroke Scale score, ranging from 0 to 42 points, with higher scores indicating more severe neurological damage), ADL score (i.e. Activities of Daily Living score, ranging from 0 to 100 points, with higher scores indicating better daily living abilities), CRP level (i.e. C-Reactive Protein, a non-specific inflammatory marker), tumor necrosis factor-alpha level (TNF-α), plasma viscosity, homocysteine level, and incidence of adverse reactions rate.

**2.3.2 Exclusion criteria.** 1. Animal experiments, case reports, and expert experience reports; 2. Duplicate publications; 3. Studies for which full-text access was not available.

### 2.4 Data extraction

Two researchers independently searched the literature, imported the search results into End-Note X9 software, removed duplicate articles, and reviewed titles and abstracts based on inclusion and exclusion criteria. The two researchers independently extracted data from the finally included literature using a unified extraction form. Extracted data included the authors, year

of publication, sample size, gender, average age, interventions, outcome measures, incidence, co-morbidities, risk factors, and treatment duration. In cases of disagreement, the issue was resolved through discussion or by seeking assistance from a third researcher.

## 2.5 Risk of bias assessment

The bias risk of the included studies was assessed based on the Cochrane Handbook. The assessment criteria included random sequence generation, allocation concealment, blinding of participants and personnel, blinding of outcome assessment, incomplete outcome data, selective reporting, and other biases. The risk level was categorized as low risk, uncertain risk, or high risk.

## 2.6 Statistical analysis

Statistical analysis was performed using RevMan 5.3 software. The efficacy rate and incidence of adverse reactions rate were analyzed as binary variables. The analysis statistic used was the risk ratio (RR), and their 95% confidence intervals (CI) were calculated. Continuous variables, including NIHSS score, ADL score, CRP, TNF-$\alpha$, plasma viscosity, and homocysteine levels, were analyzed using standardized mean difference (SMD), and their 95% CI were calculated. RR was analyzed using the Z-test. P< 0.05 indicated a statistical difference in the evaluation indicators between the two groups; otherwise, there was no statistical difference. Heterogeneity analysis was performed using a chi-square test. If statistical heterogeneity was present (P< 0.1, I^2> 50%), a random-effects model was used for analysis; otherwise, a fixed-effects model was used. Funnel plots and Egger's test were used for publication bias analysis.

# 3. Results

## 3.1 Basic characteristics of included studies

A total of 69 relevant articles were obtained through computer searches. As shown in Fig 1, the literature was screened according to the specified process, and finally, 25 studies that met the criteria were included for meta-analysis [11–35]. These studies involved a total of 2362 patients (experimental group = 1182, control group = 1180), with 1252 male and 990 female patients (JIAO J et al.'s study did not describe the gender distribution). The average age ranged from 50 to 67 years. Among these studies, 8 studies involving 786 cases were first-onset, while the remaining studies did not describe the number of disease occurrences. The majority of patients were admitted to the hospital within 24 hours of disease onset. 12 studies involving 909 patients had comorbidities such as hypertension (389), coronary heart disease (93), atrial fibrillation (56), heart failure (21), diabetes (213), or hyperlipidemia (137). The basic information of the included studies is presented in Table 1 (Detailed demographic and clinical data can be found in S1 Table).

## 3.2 Quality evaluation results of included studies

All 25 included studies were written in Chinese. 12 studies used random number table methods for grouping [13,14,18,19,21,24,27,29–33], while 5 studies mentioned random grouping without describing the specific method [16,20,22,23,26]. 1 study used a double-blind envelope method [15], 1 study used a balloting method [34], and 1 study used dice rolling [28]. Additionally, there were 5 case-control studies [11,12,17,25,35]. All reported studies had relatively complete outcome data, with no missing outcome data and no reported bias from other sources. The quality of the studies was moderate to low. The evaluation results of the risk of bias in the included studies are shown in Fig 2.

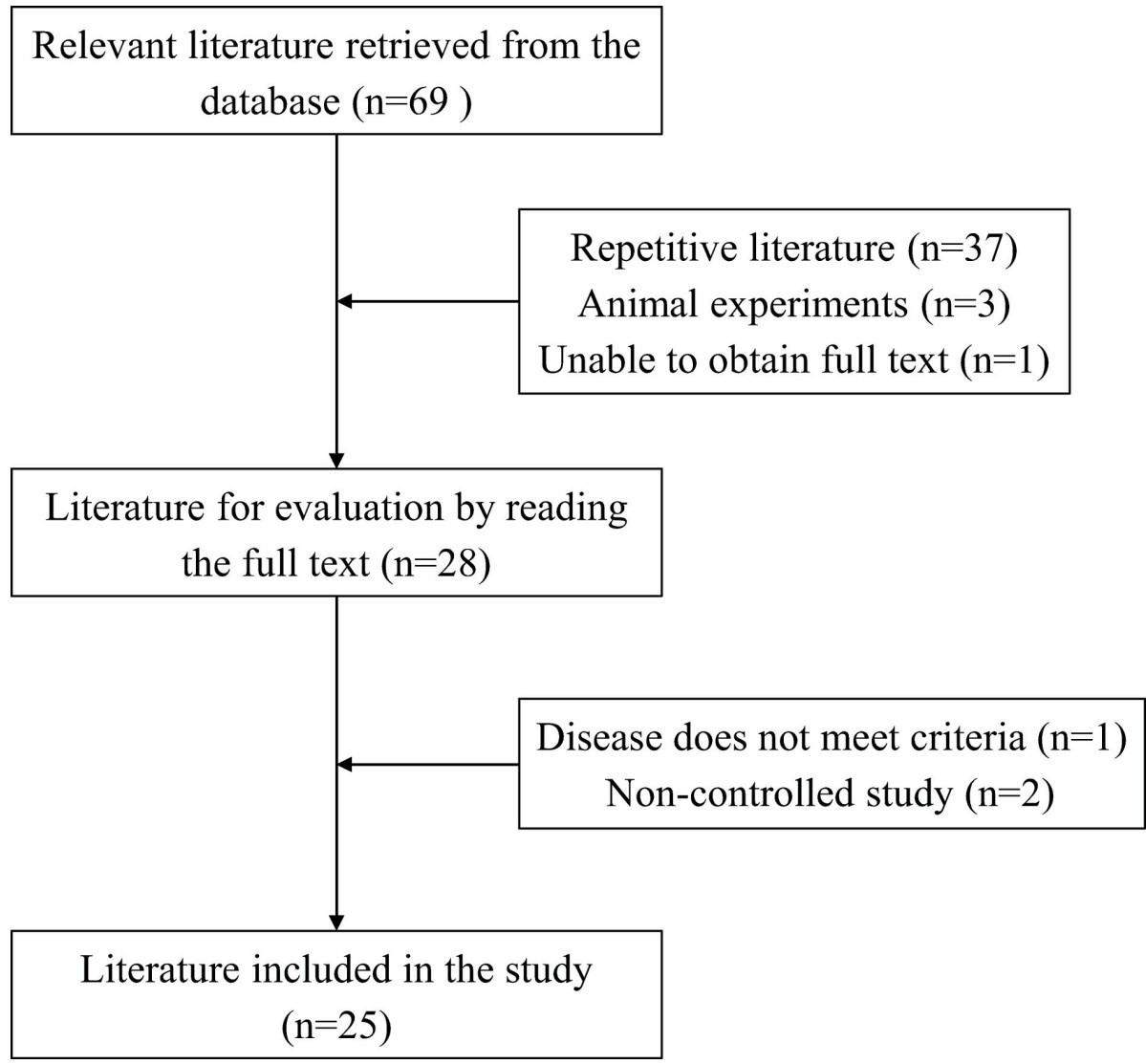

**Fig 1. Summary of the process for identifying candidate studies.**

### 3.3 Meta-analysis results

**3.3.1 Effective rate.**   A total of 19 studies [11,13,17–20,22–24,26–35] examined efficacy, and there was no statistical heterogeneity among the studies (P = 0.68, I2 = 0%), thus a fixed-effects model was used for analysis. The analysis results showed that compared to the control group, the total effective rate of the experimental group patients significantly increased [RR = 1.21, 95% CI (1.16, 1.26), P< 0.00001]. The results are shown in Fig 3.

**3.3.2 NIHSS score.**   A total of 21 studies [11–15,17–21,23–33] investigated the degree of neurological functional deficits using the NIHSS score (ranging from 0 to 42 points, with higher scores indicating more severe neurological damage). There was significant heterogeneity among the studies (P< 0.00001, I2 = 94%); therefore, a random-effects model was used for the meta-analysis. The results showed that compared to the control group, patients in the experimental group had a decrease in NIHSS scores [SMD = -1.59, 95% CI (-2.00, -1.18), P< 0.00001]. The results are shown in Fig 4.

**Table 1. The characteristics of baseline data.**

| Study (years) | Group | Number | Men | Women | Age (years) | Age (years) mean ± sd | Treatments | Outcome index |
|---|---|---|---|---|---|---|---|---|
| CHEN ZZ 2021 | E | 44 | 23 | 21 | 52–74 | 63.21±2.32 | GMI (25mg, qd, 28d) + BSC (200 mg, tid, 28d) | 1. Traditional Chinese Medicine Syndrome Points 2. NIHSS score |
| | C | 44 | 22 | 22 | 53–76 | 64.15 ±2.41 | BSC (200mg, tid, 28d) | |
| FAN QQ 2018 | E | 43 | 24 | 19 | 41–77 | 55.3±9.7 | GMI (25mg, qd, 7-14d) + BSC (200mg, tid, 20) | 1. Effective rate 2. Hemodynamic indicators 3. Adverse reactions |
| | C | 43 | 25 | 18 | 42–78 | 56.1±9.4 | GMI (25mg, qd, 20) | |
| GUO CL 2020 | E | 55 | 26 | 29 | — | 56.12±5.17 | GMI (25mg, bid, 14d) + BI (25mg, bid, 14d) | 1. Effective rate 2. NIHSS/ ADLscore 3. NSE and homocysteine levels 4. Adverse reactions |
| | C | 55 | 29 | 26 | — | 57.17±5.41 | BI (25mg, bid, 14d) | |
| GUO W 2022 | E | 42 | 22 | 20 | 39–74 | 58.32±10.83 | GMI (25mg, qd, 14d) + BI (25mg, qd, 14d) | 1. Effective rate 2. NIHSS score 3. Oxidative stress factors 3. Inflammatory factors 4. Adverse reactions |
| | C | 43 | 24 | 19 | 38–75 | 58.40±10.77 | BI (25mg, qd, 14d) | |
| HOU ZB 2020 | E | 60 | 31 | 29 | 48–81 | 65.37±10.69 | GMI (25mg, qd, 14d) + BI (25mg, bid, 14d) | 1. Effective rate 2. NIHSS score 3. Adverse reactions |
| | C | 60 | 32 | 28 | 49–80 | 65.08±10.41 | BI (25mg, bid, 14d) | |
| HU JP 2021 | E | 46 | 35 | 11 | — | 62.27±3.68 | GMI (25mg, qd, 28d) + BSC (60mg, bid, 28d) | 1. Effective rate 2. NIHSS/Fugl-Meyer score 3. Barthel index 4. GABA/Gly/ homocysteine levels 5. Content of neurotransmitters and cytokines |
| | C | 46 | 37 | 9 | — | 62.35±3.59 | BSC (60mg, bid, 28d) | |
| JIAO J 2023 | E | 30 | — | — | 57–67 | 61.3±2.1 | GMI (25mg, qd, 14d) + BI (25mg, bid, 14d) | 1. Effective rate 2. NIHSS/quality of life score 3. Inflammatory factors |
| | C | 30 | — | — | 58–68 | 61.1±1.8 | BI (25mg, bid, 14d) | |
| LIANG CC 2022 | E | 35 | 20 | 15 | 45–92 | 63.91±1.23 | GMI (25mg, qd, 14d) + BI (25mg, bid, 14d) | 1. Effective rate 2. NIHSS score 3. Hemorheology index 4. Adverse reactions |
| | C | 35 | 19 | 16 | 45–92 | 63.92±1.21 | BI (25mg, bid, 14d) | |
| LI NJ 2020 | E | 51 | 28 | 23 | 31–78 | 56.32±6.32 | GMI (30mg, bid, 7d) + BI (25mg, bid, 7d) | 1. Effective rate 2. NIHSS score 3. Barthel index 4. BNP and D-dimer levels 5. Adverse reactions |
| | C | 51 | 29 | 22 | 32–79 | 56.40±6.35 | BI (25mg, bid, 7d) | |
| LIU BC 2021 | E | 40 | 20 | 20 | 40–68 | 53.21±1.58 | GMI (25mg, qd, 14d) + BI (25mg, bid, 60d) | 1. Effective rate 2. ALD/ESS score 3. Adverse reactions |
| | C | 40 | 21 | 19 | 40–69 | 53.19±1.62 | BI (25mg, bid, 60d) | |
| LIU J 2019 | E | 34 | 19 | 15 | 46–73 | 57.9±4.3 | GMI (25mg, bid, 14d) + BI (25mg, bid, 14d) | 1. Effective rate 2. NIHSS score 3. Adverse reactions |
| | C | 34 | 16 | 18 | 43–75 | 56.3±5.4 | BI (25mg, bid, 14d) | |
| LIU Y 2021 | E | 30 | 16 | 14 | 49–87 | 68.3±9.7 | GMI (25mg, qd, 14d) + BSC (25mg, bid,14d) | 1. NIHSS/mRS score |
| | C | 30 | 16 | 14 | 48–85 | 66.4±11.5 | GMI (25mg, qd, 14d) | |
| LI X 2021 | E | 36 | 22 | 14 | — | 50.5±4.7 | GMI (25mg, qd, 14d) + BI (25mg, bid, 14d) | 1. Effective rate 2. NIHSS score |
| | C | 36 | 20 | 16 | — | 50.3±4.9 | BI (25mg, bid,14d) | |
| LUO W 2022 | E | 64 | 35 | 29 | 48–76 | 64.3±7.9 | GMI (25mg, qd, 14d) + BSC (200mg, tid, 20d) | 1. Effective rate 2. NIHSS/ADL score 3. Inflammatory factors 4. Adverse reactions |
| | C | 63 | 33 | 30 | 49–78 | 64.5±8.7 | BSC (200mg, tid, 20d) | |
| QIN RF 2020 | E | 60 | 32 | 28 | 48–82 | 67.39±2.53 | GMI (25mg, qd, 14d) + BI (25mg, bid, 14d) | 1. Effective rate 2. NIHSSscore |
| | C | 60 | 31 | 29 | 49–81 | 67.75±2.35 | BI (25mg, bid, 14d) | |

*(Continued)*

**Table 1.** (Continued)

| Study (years) | Group | Number | Men | Women | Age (years) | mean ± sd | Treatments | Outcome index |
|---|---|---|---|---|---|---|---|---|
| QIU XL 2022 | E | 53 | 27 | 26 | 46–80 | 63.93±2.25 | GMI (25mg, qd, 14d) + BI (25mgl, bid, 14d) | 1. Effective rate<br>2. NIHSS/ESS/MMSE/ADL score<br>3. Clinical indicators<br>4. Adverse reactions |
| | C | 53 | 30 | 23 | 45–78 | 63.47±2.19 | BI (25mg, bid, 14d) | |
| SHI XJ 2021 | E | 63 | 32 | 31 | 45–81 | 65.13±7.29 | GMI (25mg, qd, 14d) + BI (25mg, bid, 14d) | 1. Effective rate<br>2. NIHSS score<br>3. Barthel index<br>4. Hemorheology indicators<br>5. Number of lateral Cyclic number<br>6. Classification of collateral circulation |
| | C | 63 | 35 | 28 | 45–79 | 64.23±6.18 | BI (25mg, bid, 14d) | |
| SONG CY 2022 | E | 60 | 32 | 28 | 46–62 | 53.69±2.83 | GMI (25mg, qd, 14d) + BSC (200mg, tid, 14d) | 1. NIHSS score<br>2. Oxidative stress factors<br>3. Inflammatory factors |
| | C | 60 | 35 | 25 | 45–62 | 53.56±2.81 | BSC (200mg, tid, 14d) | |
| SONG YQ 2021 | E | 52 | 30 | 22 | 53–78 | 63.55±3.21 | GMI (25mg, qd, 14d) + BI (25mg, bid, 14d) | 1. Effective rate<br>2. NIHSS score<br>3. Hemorheology index<br>4. Adverse reactions |
| | C | 52 | 31 | 21 | 55–78 | 63.52±3.18 | BI (25mg, bid, 14d) | |
| WENG GM 2019 | E | 30 | 17 | 13 | 42–75 | 63.55±11.82 | GMI (25mg, qd, 90d) + BSC (200mg, tid, 90d) | 1. Effective rate<br>2. NIHSS score<br>3. Adverse reactions |
| | C | 30 | 18 | 12 | 43–78 | 63.49±11.92 | BSC(200mg, tid, 90d) | |
| XIAO H 2020 | E | 41 | 28 | 13 | 42–85 | 62.15±6.45 | GMI (25mg, qd, 14d) + BI (25mg, bid, 14d) | 1. Effective rate<br>2. NIHSS score<br>3. Coagulation function<br>4. homocysteine and D-dimer levels<br>5. Prognosis |
| | C | 41 | 29 | 12 | 39–80 | 61.37±6.44 | BI (25mg, bid, 14d) | |
| YANG ZM 2020 | E | 48 | 25 | 23 | 20–76 | 58.65±2.16 | GMI (25mg, qd, 60d) + BI (25mg, bid, 60d) | 1. Effective rate<br>2. NIHSS/ADL score<br>3. Adverse reactions |
| | C | 48 | 22 | 26 | 20–76 | 58.61±2.13 | BI (25mg, bid, 60d) | |
| ZHANG H 2022 | E | 40 | 28 | 12 | — | 58.42±5.41 | GMI (25mg, qd, 14d) + BSC (200mg, tid, 14d) | 1. Effective rate<br>2. NIHSS/ADL/MMSE score<br>3. Blood monitoring indicators<br>4. Adverse reactions |
| | C | 40 | 26 | 14 | — | 57.15±7.21 | BSC (200mg, tid, 14d) | |
| ZHANG Y 2022 | E | 49 | 27 | 22 | 46–75 | 62.7±5.5 | GMI (25mg, qd, 7-14d) + BSC (200mg, tid, 20d) | 1. NIHSS score<br>2. Hemodynamic related indicators<br>3. Inflammatory factors<br>4. Adverse reactions<br>5. Late ecurernce |
| | C | 49 | 26 | 23 | 47–73 | 61.3±5.2 | BSC (200mg, tid, 20d) | |
| ZHENG WW 2018 | E | 45 | 23 | 22 | — | 57.43±7.29 | GMI (25mg, bid, 14d) + BI (25mg, bid, 60d) | 1. Effective rate<br>2. ADL/ESS score<br>3. Adverse reactions |
| | C | 45 | 24 | 21 | — | 57.31±7.21 | BI (25mg, bid, 60d) | |

**Explanation of Acronyms:** E: Experimental group; C: Control group; GMI: Ginkgolide Meglumine Injection; BI: Butylphthalide Injection; BSC: Butylphthalide Soft Capsules; NIHSS score: National Institutes of Health Stroke Scale score, ranging from 0 to 42 points, with higher scores indicating more severe neurological damage; ADL score: Activities of Daily Living score, ranging from 0 to 100 points, with higher scores indicating better daily living abilities; CRP: C-Reactive Protein, a non-specific inflammatory marker; TNF-α: Tumor necrosis factor-alpha; NSE: Neuron-specific enolase; mRS score: The modified Rankin score scale, used to evaluate the neurological recovery status of stroke patients, divided into seven levels, with lower scores indicating better neurological function; ESS score: Epworth sleepiness scale; MMSE score: Mini-mental State Examination; BNP: Brain natriuretic peptide.

**3.3.3 ADL score.** A total of 6 studies [13,15,17,22,31,34] investigated the daily living abilities of post-treatment AIS patients using the ADL scale (ranging from 0 to 100 points, with higher scores indicating better daily living abilities). There was significant heterogeneity among the studies (P< 0.00001, I2 = 89%); therefore, a random-effects model was used for the meta-analysis. The results showed that compared to the control group, patients in the

A

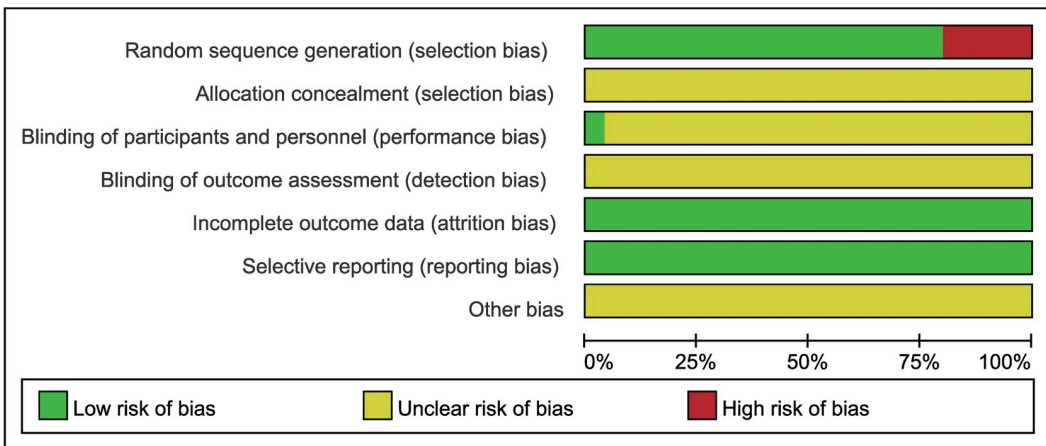

B

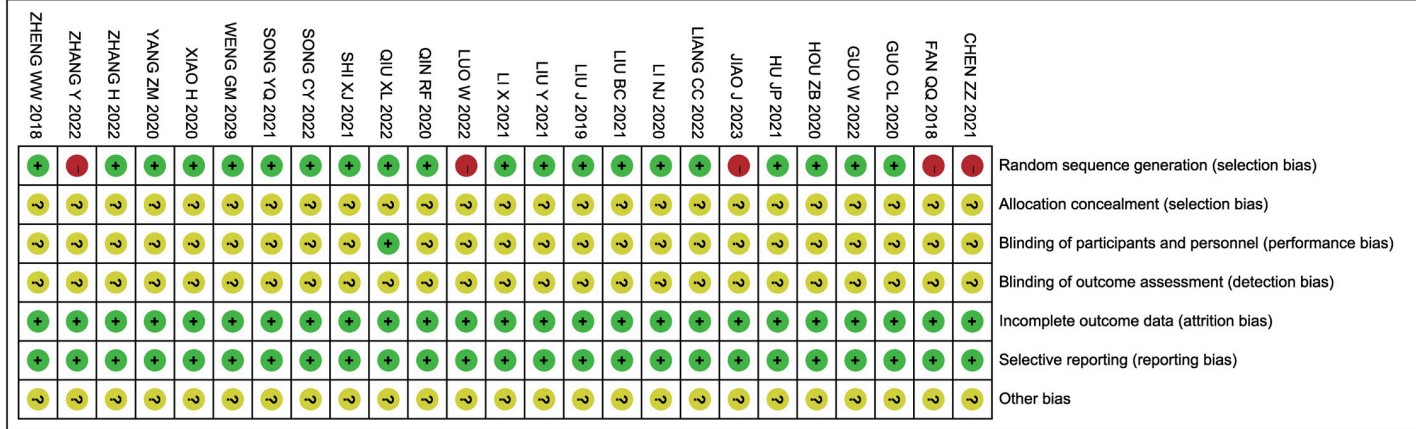

**Fig 2. Bias risk assessment of included studies.** (A): Risk of bias graph indicating the review authors' rating regarding the risk of bias, presented aspercentages, across all of the included studies; (B): Risk of bias summary indicating the review authors' judgments oneach risk of bias item for each included study.

experimental group had a significant increase in ADL scores [SMD = 2.12, 95% CI (1.52, 2.72), P< 0.00001]. The results are shown in Fig 5.

**3.3.4 C-reactive protein.** A total of 5 studies [12–14,17,18] investigated the levels of C-reactive protein (CRP), and there was significant heterogeneity among the studies (P< 0.00001, I2 = 96%); therefore, a random-effects model was used for the meta-analysis. The results showed that compared to the control group, patients in the experimental group had a significant decrease in CRP levels [SMD = -2.24, 95% CI (-3.31, -1.18), P< 0.0001]. The results are shown in Fig 6.

**3.3.5 TNF-α levels.** A total of 4 studies [11,14,17,24] investigated the levels of tumor necrosis factor-alpha (TNF-α) in post-treatment patients, and there was significant heterogeneity among the studies (P< 0.00001, I2 = 98%); therefore, a random-effects model was used for the meta-analysis. The results showed that compared to the control group, patients in the experimental group had a significant decrease in TNF-α levels [SMD = -2.74, 95% CI (-4.45, -1.03), P< 0.005]. The results are shown in Fig 7.

**3.3.6 Plasma viscosity.** A total of 4 studies [12,16,19,20] investigated plasma viscosity, and there was no significant heterogeneity among the studies (P = 0.21, I2 = 34%); therefore, a fixed-effects model was used for the meta-analysis. The results showed that compared to the

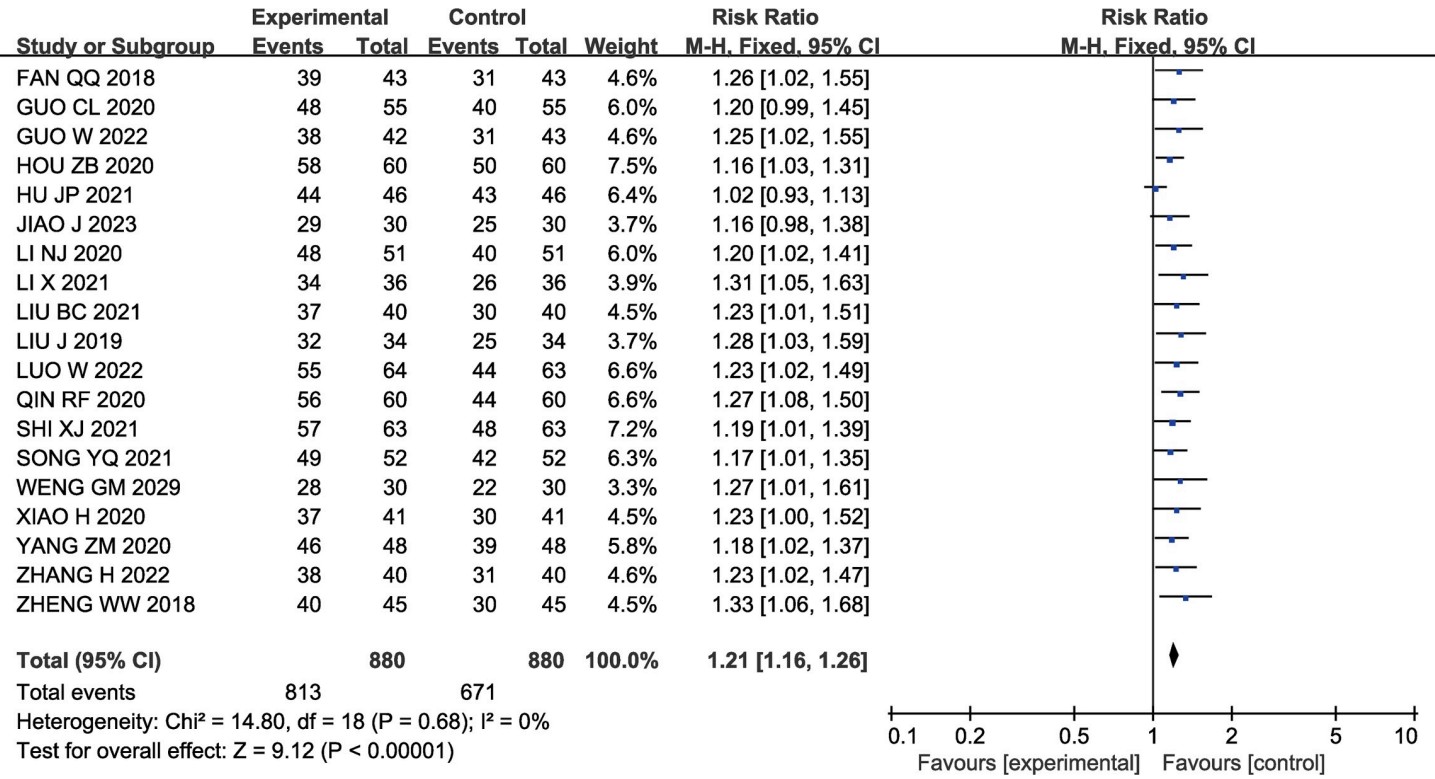

**Fig 3. Forest plot of effects of GMI combined with Butylphthalide in the treatment of AIS on effective rate.**

**Fig 4. Forest plot of NIHSS score of GMI combined with Butylphthalide in the treatment of AIS.**

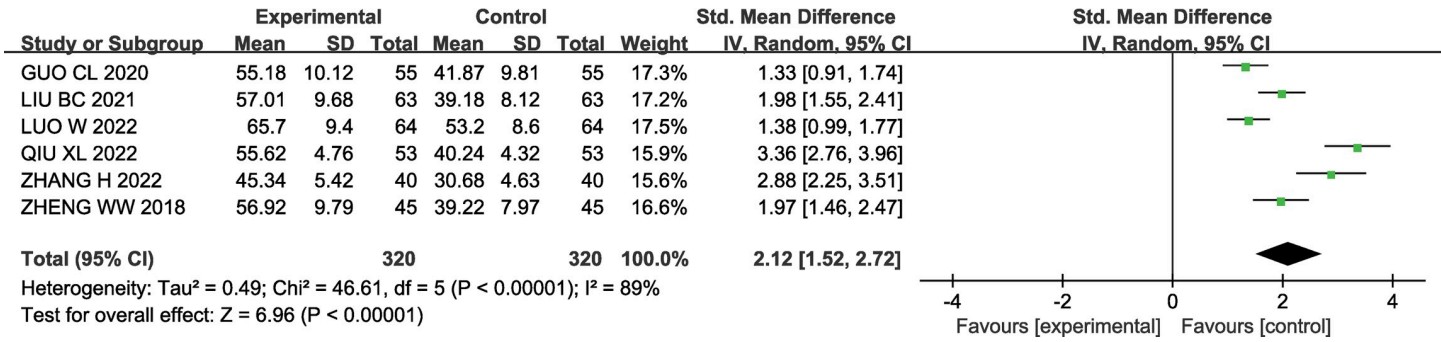

**Fig 5. Forest plot of ADL score of GMI combined with Butylphthalide in the treatment of AIS.**

control group, patients in the experimental group had a significant decrease in plasma viscosity [SMD = -0.86, 95% CI (-1.07, -0.66), P< 0.00001]. The results are shown in Fig 8.

**3.3.7 Blood homocysteine level.** A total of 4 studies [13,24,27,31] investigated the levels of homocysteine in post-treatment patients, and there was significant heterogeneity among the studies (P< 0.00001, I2 = 99%); therefore, a random-effects model was used for the meta-analysis. The results showed no significant difference in homocysteine levels between the experimental group and the control group [SMD = -0.65, 95% CI (-3.13, 1.83), P = 0.61]. The results are shown in Fig 9.

**3.3.8 Adverse reactions rate.** A total of 14 studies [12,13,15,16,19,22,26,29–35] reported relevant information on adverse events, including mild symptoms such as nausea, vomiting, gastrointestinal reactions (abdominal pain, diarrhea), abnormal liver and kidney function, coagulation abnormalities, and fever. There was no significant heterogeneity among the studies (P = 0.53, I2 = 0%); therefore, a fixed-effects model was used for the analysis. The results showed no significant difference in the incidence of adverse between the experimental group (74) and the control group (78) [SMD 0.95, 95% CI (0.71, 1.28), P = 0.73]. The results are shown in Fig 10.

**3.3.9 Sensitivity analysis.** Sensitivity analysis was conducted for all indicators, and the heterogeneity did not significantly decrease after excluding each study one by one. Except for the level of homocysteine, which showed a change in statistical significance [SMD = -2.08, 95% CI (-2.86, -1.30), P< 0.00001] (S1 Fig), there were no significant changes in the statistical differences of the other study indicators, indicating that the overall results of this meta-analysis are stable and reliable.

**3.3.10 Publication bias analysis.** Publication bias analysis was conducted using Standard Error (SE) as the vertical axis and Risk Ratio (RR) as the horizontal axis for the efficacy of the

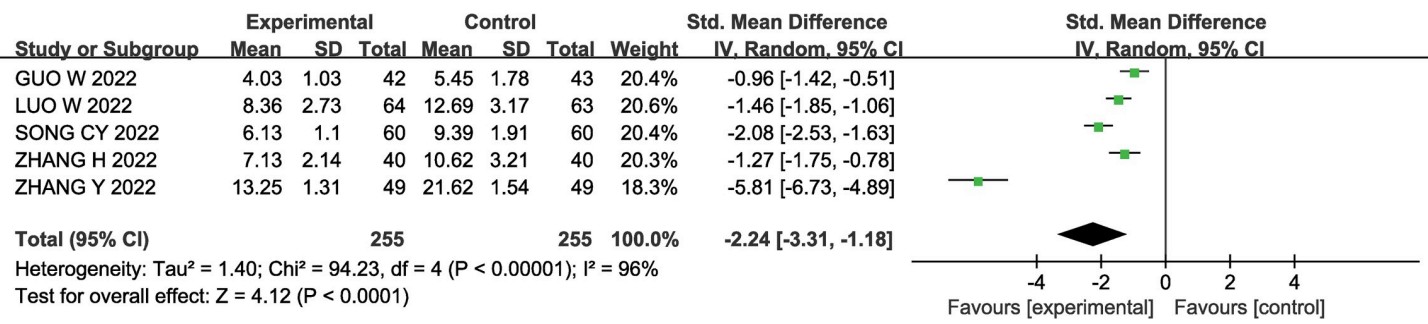

**Fig 6. Forest plot of effects of GMI combined with Butylphthalide in the treatment of AIS on CRP level.**

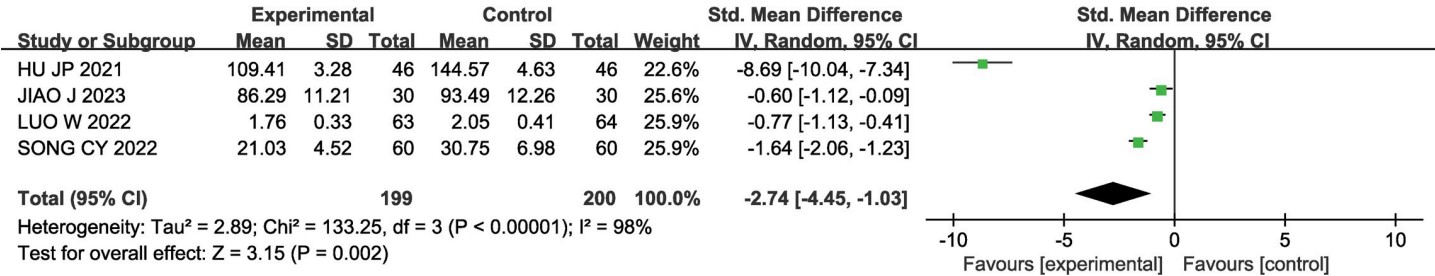

**Fig 7. Forest plot of effects of GMI combined with Butylphthalide in the treatment of AIS on TNF-α level.**

treatment. The funnel plot results showed incomplete symmetry in the distribution of included studies, indicating the possible presence of publication bias in this study. The results are shown in Fig 11. Egger's test results showed significant small-study effects (P< 0.05) in this study. Bias adjustment was performed using the trim-and-fill method, and after adding 8 additional articles, there were minimal changes in the effect values and confidence intervals before and after trimming (S2 Fig), indicating that publication bias does not affect the results of this meta-analysis.

## 4. Discussion

Acute ischemic stroke (AIS) is the most common type of stroke, accounting for approximately 70% of all strokes [1,2]. Its etiology is cerebral tissue ischemia caused by intravascular thrombosis, which can lead to severe brain tissue and neuronal damage in a short period of time. Currently, there are three major mechanisms of neuronal injury following ischemic stroke: 1) Neuronal loss caused by ischemia is one of the most direct causes of neuronal damage; 2) Overproduction of reactive oxygen species induced by ischemia exacerbates neuronal injury and results in severe functional impairment; 3) Inflammation induced by ischemia is another factor that further contributes to neuronal damage after stroke [36,37]. Therefore, the rational use of neuroprotective drugs to effectively alleviate oxidative stress and immune response may help reduce neuronal injury.

Neuroprotective therapy aims to block neuronal cell death by influencing the biochemical processes of the ischemic cascade. Currently, neuroprotective agents have shown positive results in animal models, but the negative conclusions of some clinical trials have led to doubts about the efficacy of neuroprotective therapy [38]. Traditional Chinese medicine has been widely used in China for the treatment of AIS for many years. Although studies have shown that administration of traditional Chinese medicine after ischemic stroke can improve neurological deficits in patients, a multicenter, randomized, double-blind, placebo-controlled trial

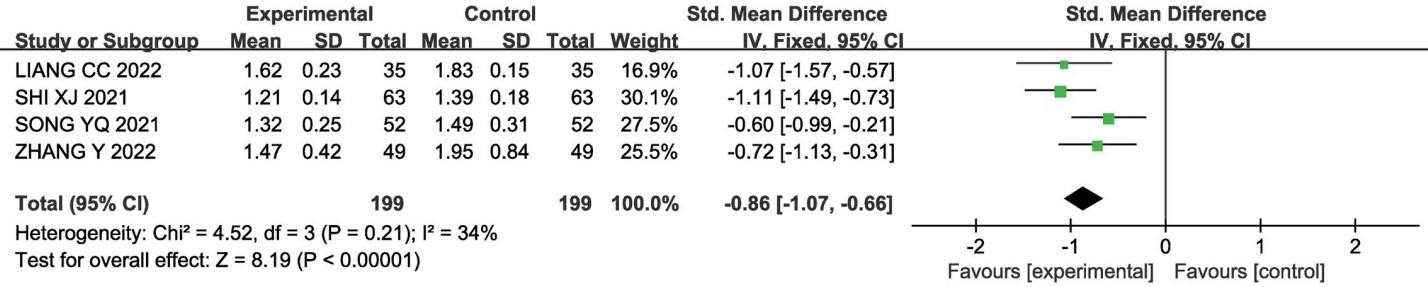

**Fig 8. Forest plot of effects of GMI combined with Butylphthalide in the treatment of AIS on plasma viscosity.**

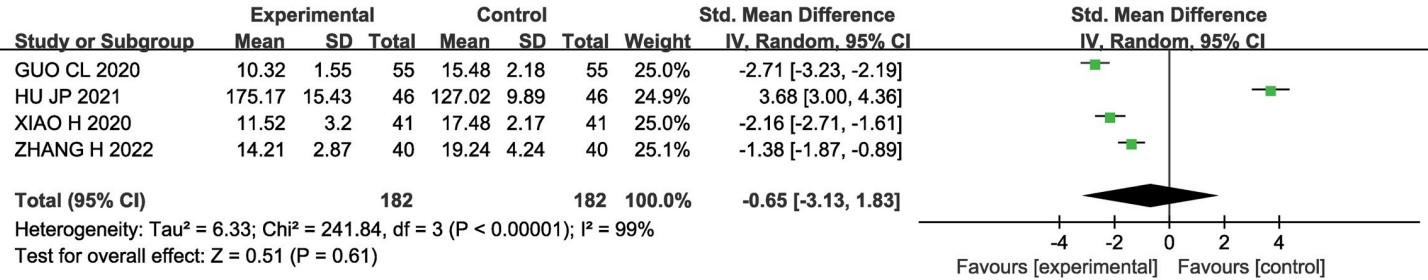

**Fig 9. Forest plot of effects of GMI combined with Butylphthalide in the treatment of AIS on Blood Homocysteine level.**

demonstrated no significant difference in the long-term outcome measure mRS score (i.e. the modified Rankin score scale, used to evaluate the neurological recovery status of stroke patients, divided into seven levels, with lower scores indicating better neurological function) [39]. This suggests that the clinical efficacy of traditional Chinese medicine needs to be further confirmed through high-quality research.

Butylphthalide is the third innovative drug with independent intellectual property rights in China and has become one of the main treatment drugs for improving neurological deficits in AIS patients. Studies have shown that Butylphthalide has a unique dual mechanism of action by reconstructing microcirculation, protecting vascular structure integrity, increasing blood flow in ischemic areas, and protecting mitochondria, enhancing ATPase activity, maintaining mitochondrial membrane stability, and reducing cell death [5–7]. However, the treatment effect of Butylphthalide on AIS is influenced by various factors, such as the severity of the patient's condition and the timing of treatment. For some ischemic strokes, Butylphthalide may not completely solve the blood flow obstruction. Clinical studies have shown that long-term use of Butylphthalide may cause adverse reactions such as headache and palpitations, and in severe cases, allergic reactions may occur. Moreover, there is a higher risk of recurrence

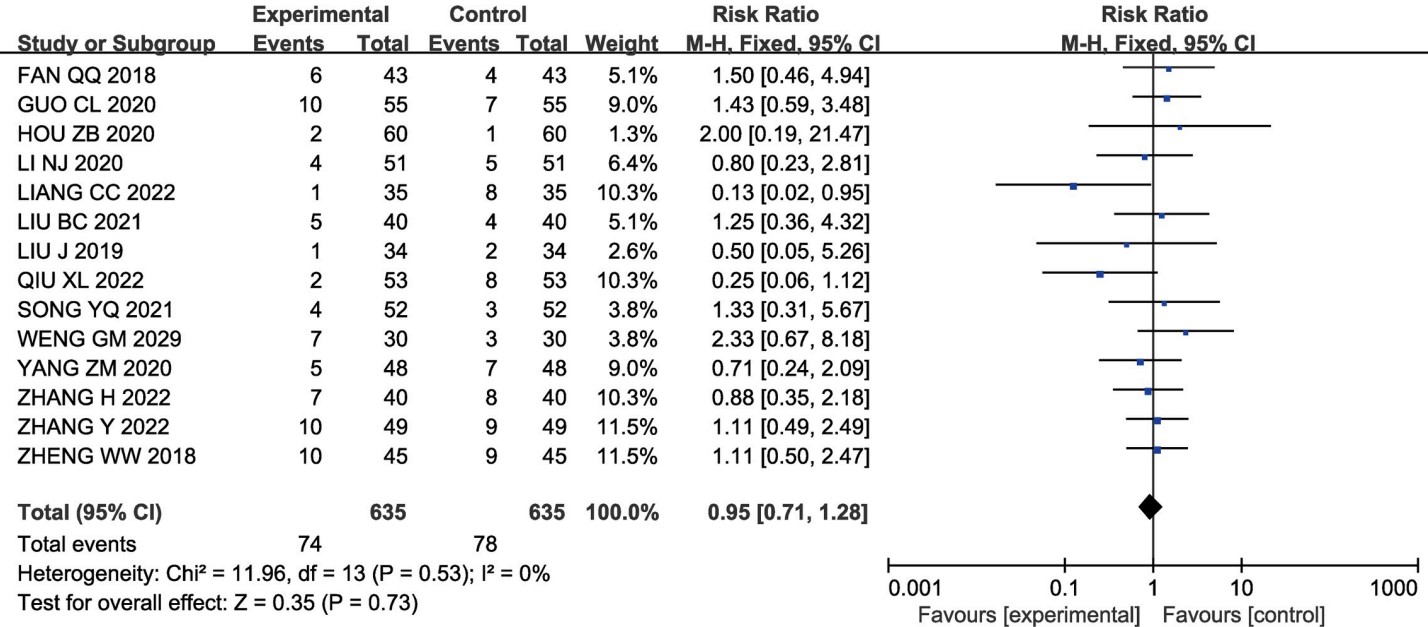

**Fig 10. Forest plot of effects of GMI combined with Butylphthalide in the treatment of AIS on Adverse reaction rate.**

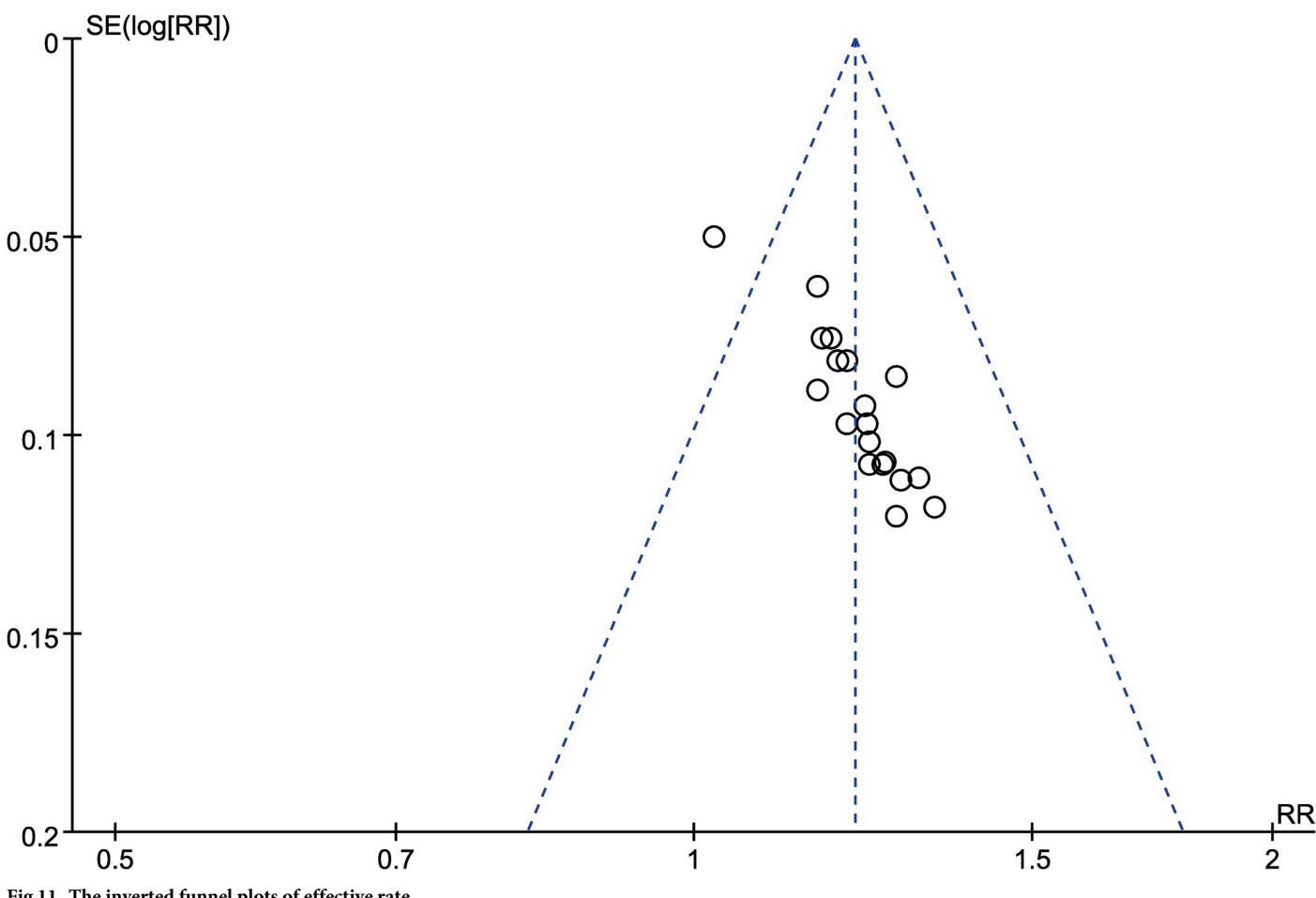

**Fig 11. The inverted funnel plots of effective rate.**

after discontinuation of the drug, which affects the treatment effect [40]. GMI is a drug composed of active ingredients extracted from Ginkgo biloba extract, which has multiple pharmacological effects, including increasing cerebral blood flow, antioxidant effects, and improving neurological function [6,8–10]. Previous studies have shown that Butylphthalide and GMI can synergistically inhibit thrombosis, increase cerebral blood flow, exert antioxidant effects, alleviate stress, improve cerebral blood supply, and protect neurons. This combination therapy can reduce thrombosis and alleviate cerebral ischemia and hypoxic conditions, thereby effectively reducing the occurrence and progression of AIS. However, there is currently no systematic review on the combination therapy of GMI and Butylphthalide.

Our research results showed that the combination of GMI and Butylphthalide significantly improved the overall effective rate in the experimental group. The National Institutes of Health Stroke Scale (NIHSS) score and the Activities of Daily Living (ADL) score are commonly used scales to assess the degree of neurological deficits and daily life activity ability in stroke diagnosis and treatment. The NIHSS score consists of 11 assessment items, including level of consciousness, ability to answer questions, ability to follow commands, eye movement, visual field, facial muscle strength, upper limb motor function, lower limb motor function, limb coordination, sensory function, articulation, and neglect. Each item is graded on scale of 3–5 points, with a score range of 0–42. A higher score indicates more severe neurological damage. ADL refers to the necessary activities performed by a person to meet daily life needs. It reflects

the most basic abilities of individuals in a home (or medical institution) and in the community. The ADL score ranges from 0–100, with a higher score indicating better daily life abilities. The scientific use of NIHSS and ADL scores can help physicians or nursing staff objectively assess the severity of stroke in patients, assisting doctors in predicting patient recovery, choosing treatment plans, evaluating the effectiveness of drug or surgical interventions, and formulating subsequent rehabilitation plans. In this analysis, the included studies had a pre-treatment NIHSS score mostly in the mild to moderate range, and the ADL score showed moderate to severe daily life functional impairment in patients. The differences in NIHSS and ADL scores between the two groups after treatment were statistically significant, indicating that the combination therapy of GMI and Butylphthalide can effectively improve the neurological dysfunction and daily living abilities of AIS patients.

Inflammatory response is an important factor contributing to further neuronal damage and functional impairment after stroke. Meta-analysis results showed that compared to the control group, patients in the experimental group had significantly reduced levels of CRP and TNF-$\alpha$, indicating that the combination of GMI and Butylphthalide can significantly alleviate the inflammatory response in AIS patients, which is consistent with previous research conclusions. Plasma viscosity is one of the indicators reflecting blood flow, and a higher plasma viscosity indicates higher whole blood viscosity and poorer blood flow dynamics. Our research results showed that the therapy significantly reduced plasma viscosity levels and improved blood flow dynamics in AIS patients. In addition, blood homocysteine level is one of the commonly used factors in clinical diagnosis and treatment. Studies have shown a positive correlation between blood homocysteine levels and the severity of ischemic stroke. However, the meta-analysis results showed no statistically significant difference in homocysteine levels between the experimental group and control group after combination therapy. To further investigate the reliability of result, we conducted sensitivity analysis and observed a statistically significant difference in homocysteine levels between the two groups after excluding the study by HU JP et al.

This result is different from our expectations, but there are reasonable explanations. First, blood homocysteine levels are influenced by various factors, including genetic factors, nutritional status, and the presence of other diseases. Although we have tried to control heterogeneity among the different studies, there may still be confounding variables resulting in differences in the results due to the presence of these underlying factors. Secondly, at the tissue and cellular level, the metabolism and degradation of blood homocysteine is a complex process that may be influenced by multiple factors, and combination therapy may not directly affect the of blood homocysteine levels. It is worth noting that sensitivity analysis is conducted to test the stability of the results, but it may not fully explain the inconsistency of the analytical results. In this study, because the number of studies and sample size included in the homocysteine level analysis is relatively small, and statistical significance was only after excluding one study, this may indicate that this study contributes less to the overall result or there may be other unknown factors causing the inconsistency with other studies. Therefore, an exact conclusion on the influence of the combination therapy of GMI and Butylphthalide on homocysteine levels in AIS patients cannot be drawn. Further research is needed to better understand the relationship between GMI, Butylphthalide, and blood homocysteine levels in stroke patients.

In terms of safety, there was no significant difference in the of adverse reactions between the experimental (n = 74) and the control group (n = 78). The main adverse events were mild to moderate symptoms, including nausea and vomiting (26 vs 32), gastrointestinal reactions such as abdominal pain and diarrhea (17 vs 13), abnormal liver and kidney function (7 vs 5), coagulation abnormalities (6 vs 11), fever (5 vs 0), elevated transaminases (3 vs 4), rash (3 vs 6), chest tightness (2 vs 1), hypotension (2 vs 1), sinus arrhythmia (1 vs 1), dizziness (1 vs 0), decreased appetite (1 vs 2), and gum bleeding (0 vs 2). It is worth noting that the number of

fever reactions in the experimental group was significantly higher than that in the control group, which may be related to the pharmacological effects of GMI and Butylphthalide. GMI has certain anti-inflammatory effects, while Butylphthalide has effects on microcirculation and antioxidant effects. The combined use of these drugs may have an impact on the temperature regulation of patients, leading to fever. In addition, factors such as the medicinal material source, climatic differences in the medicinal material planting base, harvest time, processing methods, and processes of Chinese medicine injections also affect their safety [41,42]. It should be noted that although the fever reaction did not reach significance, it may be due to the smaller sample size or other unknown factors causing random variation. Therefore, more studies are needed to further confirm and explain whether this difference has clinical significance. Overall, according to our research results, the combination of GMI and Butylphthalide shows safety in the treatment of AIS. In clinical, doctors should assess the risks and benefits of treatment based on specific situation and physical condition of the patient, and closely monitor and manage adverse reactions.

To determine if our research conclusions were by specific studies, we conducted sensitivity analysis by removing individual studies one by one. In this study, except for the statistical change in homocysteine levels, there were no significant changes in the statistical differences of the other study indicators, indicating the overall stability and reliability of this meta-analysis. Additionally, the Egger's test results showed significant small-study effects, suggesting that the research results may be more susceptible to random errors, leading to biased results. We further used the trim-and-fill method to assess publication bias, and after adding 8 additional studies, there were minimal changes in the effect values and confidence intervals before and after trimming, indicating that publication bias does not affect our research conclusions.

Although our study has achieved some positive results, there are also limitations. Firstly, the included literature did not provide detailed descriptions of specific random methods, allocation concealment, and blinding methods, which may lead to selection bias and implementation bias, and the quality of the literature needs improvement. Secondly, most of the studies in this study were single-center clinical studies with small sample sizes, and the efficacy indicators of each study lacked consistency. Therefore, further large-sample, multicenter, high-quality randomized controlled trials (RCTs) are needed to be conducted to further validate these results.

## 5. Conclusion

Based on the comprehensive results of this meta-analysis, we draw the following conclusions: the combination therapy of GMI and Butylphthalide may have certain advantages in improving neurological function and anti-inflammatory effects in patients with acute ischemic stroke. And the safety profile is favorable., but it is still necessary to closely monitor the occurrence of adverse reactions. In the future, more high-quality studies are still needed to validate the efficacy and safety of this combination therapy in practical clinical applications.

## Supporting information

**S1 Checklist. PRISMA 2020 checklist.**
(DOCX)

**S1 Fig. Forest plot of the effect of GMI combined with Butylphthalide treatment on blood homocysteine levels in AIS after excluding one study.**
(TIF)

**S2 Fig. The inverted funnel plots of publication bias in effectiveness rates using the trim and fill method.**
(TIF)

**S1 Table. The detailed characteristics of Baseline Data.**
(DOCX)

**S1 File. PRISMA 2020 flow diagram.**
(DOCX)

**S1 Data. Study's minimal underlying data: Raw data involved in meta-analysis.**
(XLSX)

## Author Contributions

**Conceptualization:** Xia Chen, Pan-Feng Feng.

**Data curation:** Jia-Qi Zhou, Xiang-Fan Chen.

**Formal analysis:** Jia-Qi Zhou.

**Funding acquisition:** Xiang-Fan Chen, Pan-Feng Feng.

**Methodology:** Xia Chen.

**Software:** Jia-Qi Zhou, Xiang-Fan Chen, Xia Chen.

**Writing – original draft:** Jia-Qi Zhou.

**Writing – review & editing:** Xia Chen, Pan-Feng Feng.

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
