## [Decision Letter · Decision Letter 0]

18 Sep 2023

PONE-D-23-22642Meta-analysis of the efficacy and safety of Ginkgolide Meglumine Injection combined with Butylphthalide in the treatment of Acute Cerebral InfarctionPLOS ONE

Dear Dr. SHUAI,

Thank you for submitting your manuscript to PLOS ONE. After careful consideration, we feel that it has merit but does not fully meet PLOS ONE’s publication criteria as it currently stands. Therefore, we invite you to submit a revised version of the manuscript that addresses the points raised during the review process.

We look forward to receiving your revised manuscript.

Kind regards,

Tariq Jamal Siddiqi

Academic Editor

PLOS ONE

Journal Requirements:

“This work was supported by Jiangsu Pharmaceutical Association-HengRui Hospital Pharmacy Fund (No. H202047), Nantong Health Commission Fund (No. QA2021007, QA2021006, QA2021014) and Development Fund of KangDa college of Nanjing medical university (No. KD2021KYJJZD127). The funders had no role in study design, data collection and analysis, decision to publish, or preparation of the manuscript. The funders had no role in study design, data collection and analysis, decision to publish, or preparation of the manuscript.”

Reviewers' comments:

Reviewer's Responses to Questions

**Comments to the Author**

1. Is the manuscript technically sound, and do the data support the conclusions?

Reviewer #1: No

Reviewer #2: Yes

2. Has the statistical analysis been performed appropriately and rigorously? 

Reviewer #1: Yes

Reviewer #2: Yes

3. Have the authors made all data underlying the findings in their manuscript fully available?

Reviewer #1: Yes

Reviewer #2: Yes

4. Is the manuscript presented in an intelligible fashion and written in standard English?

Reviewer #1: No

Reviewer #2: Yes

5. Review Comments to the Author

Reviewer #1: Thank you for submitting you article to Plos One for consideration. My comments:

In the introduction section. The authors reports efficacy of Ginkgolide Meglumine Injection and Butylpthalide in cases of Acute cerebral infarction. However, the author does not describe any mechanisms that may explain its effects.

The confidence interval in the ADL score outcome needs correction

High levels of heterogeneity were noted in some areas (I2 > 89% in all cases), and a random-effects model was used. High heterogeneity may signify inconsistencies across the studies, possibly due to different study designs, patient characteristics, or measurement techniques. Additional analyses, such as subgroup analyses or sensitivity analyses, could help in understanding the sources of heterogeneity.

The mention of publication bias analysis and the results' symmetry is positive, but other tests (e.g., Egger's test) or further detail on how the funnel plot was assessed could strengthen this statement.

Description of adverse events outcome. Unclear how the adverse events were defined (i.e. life-threatening, anaphylaxis, allergy)

The discussion could provide more information on how these findings fit within the broader context of existing research and literature in the field of ischemic stroke treatment.

Reviewer #2: Li et al. conducted a meta-analysis of the efficacy and safety of Ginkgolide Meglumine Injection combined with Butylphthalide in the treatment of Acute Cerebral Infarction and concluded that the combination therapy of GMI and Butylphthalide appears promising for the treatment of ACI, offering potential benefits in terms of improved neurological function and daily living abilities. This meta-analysis provides valuable insights, however, addressing the following suggestions will help improve the quality of the manuscript.

1. While the introduction provides a brief overview of the topic, incorporating more recent statistics and references regarding the burden of the disease will highlight the significance of this study and help readers grasp the current context and relevance of the research.

2. Consider presenting the methodology in a continuous narrative form. This change will enhance the flow and readability of the methods section, making it easier for readers to follow the study's design and execution.

3. Abbreviate NHISS (National Institutes of Health Stroke Scale) and ADL (Activities of Daily Living) and provide a brief description of these scores for better comprehensibility. Clarifying the meaning of these abbreviations will aid readers in understanding the significance of these assessments.

4. In the methods section, specify how duplicates were removed from the dataset. This information is critical for transparency and reproducibility, allowing readers to understand the data cleaning process.

5. In the section on basic baseline characteristics, provide a brief description of the characteristics of the study cohorts. This will offer readers a clearer picture of the study population, aiding in the interpretation of the results.

6. Enhance the baseline characteristics table by including key demographic and clinical data, such as age, gender distribution, comorbidities (e.g., hypertension, diabetes), time to treatment, baseline neurological status (NIHSS scores), and relevant risk factors (e.g., smoking, alcohol use). This addition will improve the generalizability of the findings, enable subgroup analyses, and aid in assessing the impact of baseline characteristics on treatment outcomes.

7. In the quality evaluation section, summarize the overall quality of the included studies. This summary will help readers quickly assess the reliability and validity of the evidence presented in the meta-analysis.

8. In the results section, ensure that numerical results are provided for homocysteine levels and the adverse reaction rate. Quantitative data will allow readers to understand the magnitude of the findings.

9. In the publication bias analysis, use the expanded form of SE (Standard Error) for better comprehensibility. This change will ensure that readers easily grasp the terminology used in the analysis.

10. In the discussion, expand on the clinical implications of your findings, particularly regarding how the observed improvements in NIHSS scores and ADL scores may translate to patient outcomes and clinical practice. Providing this insight will help bridge the gap between research findings and their real-world impact.

11. Overall, the manuscript has many abbreviation, language and grammatical errors and needs thorough proofreading. Ensuring the manuscript's language is polished and error-free enhances the professionalism and readability of the research.

12. Maintain consistency in the use of terminology and abbreviations throughout the manuscript. This consistency improves clarity and reduces potential confusion for readers.

13. The authors should include a conclusion section summarizing the main findings and their practical implications to provide a clear and concise takeaway for readers. A well-crafted conclusion enhances the manuscript's overall impact by summarizing key takeaways and emphasizing the study's significance.

6. PLOS authors have the option to publish the peer review history of their article (what does this mean?). If published, this will include your full peer review and any attached files.

Reviewer #1: **Yes: **Mahammed Z Khan suheb

Reviewer #2: No

---

## [Author Response · Author response to Decision Letter 0]

6 Oct 2023

Dear Editors and Reviewers:

Thank you for your letter and for the reviewers’ comments concerning our manuscript entitled “Meta-analysis of the efficacy and safety of Ginkgolide Meglumine Injection combined with Butylphthalide in the treatment of Acute Cerebral Infarction” (ID: PONE-D-23-22642). Those comments are all valuable and very helpful for revising and improving our paper, as well as the important guiding significance to our researches. We have studied comments carefully and have made correction which we hope meet with approval. Revised portion are marked in blue in the paper. The main corrections in the paper and the responds to the reviewer’s comments are as flowing:

Reviewer #1:

1. In the introduction section. The authors reports efficacy of Ginkgolide Meglumine Injection and Butylpthalide in cases of Acute cerebral infarction. However, the author does not describe any mechanisms that may explain its effects.

Author's response to comments: Thank you for your valuable comments. We have supplemented and elaborated on the pharmacological mechanisms of Ginkgolide Meglumine Injection combined with Butylphthalide in the treatment of Acute Ischemic Stroke in the third and fourth paragraphs of the introduction. The chemical name of Butylphthalide is racemic 3-phenylbutan-2-one, which is characterized by high lipid solubility, easy passage through the blood-brain barrier, direct action on the infarct site, rapid onset, and significant effects. Numerous studies have shown that Butylphthalide inhibits neuronal apoptosis by blocking the cascade reaction of caspases. Additionally, Butylphthalide significantly improves neuronal cell function by protecting mitochondria, alleviating inflammatory response, and enhancing microcirculation, with no serious adverse reactions. The active components of Ginkgolide Meglumine, namely Ginkgolide K, B, and A, effectively antagonize platelet aggregation, improve brain edema, and can also reduce NF-κB bioactivity by inhibiting the cellular mitochondrial apoptosis pathway, thereby alleviating neuronal apoptosis and inflammatory response, protecting neuronal cells, and promoting neurological function recovery.

2. The confidence interval in the ADL score outcome needs correction.

Author's response to comments: Thank you for pointing out this error. We have made corrections to the confidence intervals of the ADL scores, as detailed in the research results section "3.3.3 ADL score". Furthermore, we have reviewed the description of confidence intervals for all results to ensure there are no similar errors.

3. High levels of heterogeneity were noted in some areas (I2 > 89% in all cases), and a random-effects model was used. High heterogeneity may signify inconsistencies across the studies, possibly due to different study designs, patient characteristics, or measurement techniques. Additional analyses, such as subgroup analyses or sensitivity analyses, could help in understanding the sources of heterogeneity.

Author's response to comments: Thank you very much for the valuable comments from the reviewer. Our study does indeed exhibit high heterogeneity in certain aspects (I2 > 89% in all cases). Therefore, in accordance with the reviewer's suggestions, we conducted sensitivity analyses using the the method of one by one elimination of references to examine all outcome measures. The results showed that, upon removing one study, there was a statistically significant change in homocysteine levels, while the statistical differences in the remaining outcome measures did not show significant changes, indicating that the overall results of this meta-analysis are stable and reliable. We have provided a detailed explanation of the possible reasons for the statistical change in homocysteine levels in the discussion section. We are very grateful for your guidance and support.

4. The mention of publication bias analysis and the results' symmetry is positive, but other tests (e.g., Egger's test) or further detail on how the funnel plot was assessed could strengthen this statement.

Author's response to comments: The assessment of funnel plot is subjective and cannot quantitatively detect publication bias, and the reviewer's comment on this is very important. Following the reviewer's comment, we performed Egger's test, and the results indicated a potential presence of significant small-study effects. Additionally, we further evaluated publication bias using the trim-and-fill method. After adding 8 supplementary studies, the changes in effect sizes and confidence intervals before and after trimming were minimal (see Supplementary Fig 1), suggesting that publication bias does not affect the overall conclusion of this meta-analysis.

5. Description of adverse events outcome. Unclear how the adverse events were defined (i.e. life-threatening, anaphylaxis, allergy).

Author's response to comments: All adverse events involved in this study were mild to moderate, mainly including nausea and vomiting (26 vs 32), gastrointestinal reactions such as abdominal pain and diarrhea (17 vs 13), abnormal liver and kidney function (7 vs 5), coagulation abnormalities (6 vs 11), fever (5 vs 0), elevated transaminases (3 vs 4), rash (3 vs 6), chest tightness (2 vs 1), hypotension (2 vs 1), sinus arrhythmia (1 vs 1), dizziness (1 vs 0), decreased appetite (1 vs 2), and gum bleeding (0 vs 2). We have provided a detailed description of the types and incidence rates of adverse reactions in the results section "3.3.8 Adverse reactions rate" and the “Discussion” section.

6. The discussion could provide more information on how these findings fit within the broader context of existing research and literature in the field of ischemic stroke treatment.

Author's response to comments: We would like to express our gratitude to the reviewer for their valuable comments. In the discussion section, we have provided a detailed account of the connections between our research findings and existing studies. We have also analyzed whether our findings support, extend, or differ from previous research, and offered potential explanations for any observed differences. For more specific information, please refer to the "Discussion" section. Thank you sincerely for your guidance and support.

Reviewer #2:

1. While the introduction provides a brief overview of the topic, incorporating more recent statistics and references regarding the burden of the disease will highlight the significance of this study and help readers grasp the current context and relevance of the research.

Author's response to comments: We appreciate the valuable comments from the reviewer. Building upon relevant studies from the Global Burden of Disease Database for 2019, in the introduction section, we have provided additional detailed description regarding the statistical data on acute ischemic stroke. As of 2019, there were a total of 12.2 million new cases of stroke globally, with 101 million prevalent cases and a stroke-related Disability-Adjusted Life Years (DALYs) of 143 million. Stroke was responsible for the death of 6.55 million individuals. From 1990 to 2019, the absolute number of new stroke cases increased by 70.0%, the number of current stroke cases increased by 85.0%, the number of deaths increased by 43.0%, and the number of DALYs caused by stroke increased by 32.0%. Globally, stroke remains the second leading cause of death (11.6% of total deaths), the third leading cause of death and disability (5.7% of total deaths). Over the past 30 years, as one of the main subtypes of stroke, the number of deaths from ischemic stroke has increased from 2.04 million to 3.29 million, and it is projected to reach 4.9 million by 2030. This has resulted in a significant economic burden and has become a severe global public health issue. The inclusion of these data indeed enhances the visual representation of the urgency for stroke treatment and the significance of conducting this study.

2. Consider presenting the methodology in a continuous narrative form. This change will enhance the flow and readability of the methods section, making it easier for readers to follow the study's design and execution.

Author's response to comments: Following the advice of the reviewer, we have provided a revised description of the research methodology in a coherent narrative format. Please refer to "2. Materials and Methods" for detailed information.

3. Abbreviate NHISS (National Institutes of Health Stroke Scale) and ADL (Activities of Daily Living) and provide a brief description of these scores for better comprehensibility. Clarifying the meaning of these abbreviations will aid readers in understanding the significance of these assessments.

Author's response to comments: The NIHSS score, also known as the National Institutes of Health Stroke Scale score, consists of 11 assessment items, including level of consciousness, ability to answer questions, ability to follow commands, eye movement, visual field, facial muscle strength, upper limb motor function, lower limb motor function, limb coordination, sensory function, articulation, and neglect. Each item is graded on scale of 3-5 points, with a score range of 0-42. A higher score indicates more severe neurological damage. The ADL score, also known as the Activities of Daily Living score, refers to the necessary activities that a person performs every day to meet the needs of daily life, reflecting the most basic abilities of people at home (or medical institutions) and in the community. The ADL score ranges from 0-100, with a higher score indicating better daily life abilities. We have provided a detailed description of the methods and significance of NIHSS and ADL score in the research methodology section "2.3.1 Inclusion criteria" as well as in the “Discussion” section.

4. In the methods section, specify how duplicates were removed from the dataset. This information is critical for transparency and reproducibility, allowing readers to understand the data cleaning process.

Author's response to comments: The literature results, independently retrieved by two researchers, were imported into EndNote X9 software. Duplicate articles were eliminated using the software, and then titles and abstracts were manually reviewed. When necessary, full-text articles were read in their entirety. The literature inclusion screening was conducted based on the inclusion and exclusion criteria.

5. In the section on basic baseline characteristics, provide a brief description of the characteristics of the study cohorts. This will offer readers a clearer picture of the study population, aiding in the interpretation of the results.

Author's response to comments: In accordance with the reviewer's suggestions, we provided a brief description of the basic characteristics of the study population, including gender, age, number of disease occurrences, time from onset to admission, and comorbidities in the baseline characteristics section. These studies involved a total of 2362 patients (experimental group=1182, control group=1180), with 1252 male and 990 female patients (JIAO J et al.'s study did not describe the gender distribution). The average age ranged from 50 to 67 years. Among these studies, 8 studies involving 786 cases were first-onset, while the remaining studies did not describe the number of disease occurrences. The majority of patients were admitted to the hospital within 24 hours of disease onset. 12 studies involving 909 patients had comorbidities such as hypertension (389), coronary heart disease (93), atrial fibrillation (56), heart failure (21), diabetes (213), or hyperlipidemia (137). For detailed information, please refer to "3.1 Basic characteristics of included studies".

6. Enhance the baseline characteristics table by including key demographic and clinical data, such as age, gender distribution, comorbidities (e.g., hypertension, diabetes), time to treatment, baseline neurological status (NIHSS scores), and relevant risk factors (e.g., smoking, alcohol use). This addition will improve the generalizability of the findings, enable subgroup analyses, and aid in assessing the impact of baseline characteristics on treatment outcomes.

Author's response to comments: Following the reviewer's suggestions, we have revised and improved the baseline characteristics table, adding age, gender, comorbidities, treatment duration, and baseline NIHSS score. Due to the large amount of information in the baseline data table, we have provided only essential baseline data in the paper. Detailed demographic and clinical data can be found in " Supplementary Table 1. The detailed characteristics of Baseline Data".

7. In the quality evaluation section, summarize the overall quality of the included studies. This summary will help readers quickly assess the reliability and validity of the evidence presented in the meta-analysis.

Author's response to comments: In the quality assessment section, we summarized the overall quality of the included studies. All 25 studies included in the analysis were Chinese literature. Among them, 20 studies used randomization, while 5 studies were case-control studies. The reporting of study outcome data was complete in all reports, with no loss of outcome data or reporting of other sources of bias. The overall quality of the included studies was considered moderate.

8. In the results section, ensure that numerical results are provided for homocysteine levels and the adverse reaction rate. Quantitative data will allow readers to understand the magnitude of the findings.

Author's response to comments: In the “Results” and “Discussion” section, we provided a detailed description of the types and numerical values of adverse reaction rates. We also discussed the statistical results and sensitivity analysis of homocysteine levels. Detailed data on homocysteine levels and adverse reaction rates can be found in the original data of the supplementary materials. All adverse events involved in this study were mild to moderate, mainly including nausea and vomiting (26 vs 32), gastrointestinal reactions such as abdominal pain and diarrhea (17 vs 13), abnormal liver and kidney function (7 vs 5), coagulation abnormalities (6 vs 11), fever (5 vs 0), elevated transaminases (3 vs 4), rash (3 vs 6), chest tightness (2 vs 1), hypotension (2 vs 1), sinus arrhythmia (1 vs 1), dizziness (1 vs 0), decreased appetite (1 vs 2), and gum bleeding (0 vs 2). 

9. In the publication bias analysis, use the expanded form of SE (Standard Error) for better comprehensibility. This change will ensure that readers easily grasp the terminology used in the analysis.

Author's response to comments: Due to the subjective nature of the assessment of funnel plots, which cannot quantitatively detect publication bias, we followed the suggestions of two reviewers and conducted an Egger's test in the analysis of publication bias. The results showed potential evidence of significant small-study effects. To further estimate publication bias, we performed a trim-and-fill analysis. After adding 8 additional studies, the changes in effect size and confidence interval before and after the trim-and-fill analysis were minimal (Supplementary Fig 2). This indicates that publication bias does not affect the overall conclusion of this meta-analysis.

10. In the discussion, expand on the clinical implications of your findings, particularly regarding how the observed improvements in NIHSS scores and ADL scores may translate to patient outcomes and clinical practice. Providing this insight will help bridge the gap between research findings and their real-world impact.

Author's response to comments: Thank you for this valuable suggestion from the reviewer. It is indeed important to expand the discussion on the clinical significance of the research results. We have further improved the content of the Discussion section and thoroughly discussed the clinical implications of all the study indicators, particularly the NIHSS and ADL scores. A higher NIHSS score indicates more severe neurological damage. ADL score reflects the most basic abilities of individuals in a home (or medical institution) and in the community. A higher ADL score indicates better daily life abilities. The scientific use of NIHSS and ADL scores can help physicians or nursing staff objectively assess the severity of stroke in patients, assisting doctors in predicting patient recovery, choosing treatment plans, evaluating the effectiveness of drug or surgical interventions, and formulating subsequent rehabilitation plans. In this analysis, the included studies had a pre-treatment NIHSS score mostly in the mild to moderate range, and the ADL score showed moderate to severe daily life functional impairment in patients. The differences in NIHSS and ADL scores between the two groups after treatment were statistically significant, indicating that the combination therapy of GMI and Butylphthalide can effectively improve the neurological dysfunction and daily living abilities of AIS patients. Please refer to the "Discussion" section for details.

11. Overall, the manuscript has many abbreviation, language and grammatical errors and needs thorough proofreading. Ensuring the manuscript's language is polished and error-free enhances the professionalism and readability of the research.

Author's response to comments: Following the reviewer's suggestions, we have thoroughly proofread the abbreviations, language, and grammar errors in this manuscript. We would like to express our gratitude to Dr. Pan-Feng Feng for his contribution to the proofreading work.

12. Maintain consistency in the use of terminology and abbreviations throughout the manuscript. This consistency improves clarity and reduces potential confusion for readers.

Author's response to comments: We have proofread the terminology and abbreviations used in this manuscript to ensure consistency throughout.

13. The authors should include a conclusion section summarizing the main findings and their practical implications to provide a clear and concise takeaway for readers. A well-crafted conclusion enhances the manuscript's overall impact by summarizing key takeaways and emphasizing the study's significance.

Author's response to comments: The reviewer's comment is highly significant. Based on the results of this meta-analysis, we have incorporated concise language to enhance the description of the conclusion section in the manuscript. The specific details are as follows: the combination therapy of GMI and Butylphthalide may have certain advantages in improving neurological function and anti-inflammatory effects in patients with acute ischemic stroke. And the safety profile is favorable., but it is still necessary to closely monitor the occurrence of adverse reactions. In the future, more high-quality studies are still needed to validate the efficacy and safety of this combination therapy in practical clinical applications. Please refer to the "5. Conclusion" section in the manuscript for detailed information.

Thank you again for the valuable feedback from the editor and reviewers on this study. We are looking forward to hearing from you.

Best regards,

Sincerely

Corresponding author:

Name: Xia-Chen & Pan-Feng Feng

E-mail: cxia66@126.com

929083891@qq.com

Address: Department of Pharmacy, Affiliated Hospital 2 of Nantong University, and First People’s Hospital of Nantong City, Nantong, Jiangsu Province, China

---

## [Decision Letter · Decision Letter 1]

14 Dec 2023

Meta-analysis of the efficacy and safety of Ginkgolide Meglumine Injection combined with Butylphthalide in the treatment of Acute Ischemic Stroke

PONE-D-23-22642R1

Dear Dr. SHUAI,

We’re pleased to inform you that your manuscript has been judged scientifically suitable for publication and will be formally accepted for publication once it meets all outstanding technical requirements.

Kind regards,

Tariq Jamal Siddiqi

Academic Editor

PLOS ONE

Additional Editor Comments (optional):

Reviewers' comments:

Reviewer's Responses to Questions

**Comments to the Author**

1. If the authors have adequately addressed your comments raised in a previous round of review and you feel that this manuscript is now acceptable for publication, you may indicate that here to bypass the “Comments to the Author” section, enter your conflict of interest statement in the “Confidential to Editor” section, and submit your "Accept" recommendation.

Reviewer #2: All comments have been addressed

Reviewer #3: All comments have been addressed

2. Is the manuscript technically sound, and do the data support the conclusions?

Reviewer #2: Yes

Reviewer #3: Yes

3. Has the statistical analysis been performed appropriately and rigorously? 

Reviewer #2: Yes

Reviewer #3: Yes

4. Have the authors made all data underlying the findings in their manuscript fully available?

Reviewer #2: Yes

Reviewer #3: Yes

5. Is the manuscript presented in an intelligible fashion and written in standard English?

Reviewer #2: Yes

Reviewer #3: Yes

6. Review Comments to the Author

Reviewer #2: The authors have diligently incorporated the suggested revisions and improvements, which have significantly strengthened the overall quality and clarity of the manuscript. Based on the comprehensive revisions, I recommend the acceptance of this manuscript for publication.

Reviewer #3: (No Response)

7. PLOS authors have the option to publish the peer review history of their article (what does this mean?). If published, this will include your full peer review and any attached files.

Reviewer #2: No

Reviewer #3: No

---

## [Editor Report · Acceptance letter]

27 Dec 2023

PONE-D-23-22642R1 

PLOS ONE

Dear Dr. SHUAI, 

I'm pleased to inform you that your manuscript has been deemed suitable for publication in PLOS ONE. Congratulations! Your manuscript is now being handed over to our production team.

Kind regards, 

on behalf of

Dr. Tariq Jamal Siddiqi 

Academic Editor

PLOS ONE